# Attitudes of Non-Disabled Pupils towards Disabled Pupils to Promote Inclusion in the Physical Education Classroom

**DOI:** 10.3390/children10061008

**Published:** 2023-06-02

**Authors:** Serafín Delgado-Gil, David Manuel Mendoza-Muñoz, Carmen Galán-Arroyo, Ángel Denche-Zamorano, Jose Carmelo Adsuar, Carlos Mañanas-Iglesias, Antonio Castillo-Paredes, Jorge Rojo-Ramos

**Affiliations:** 1Physical Activity for Education, Performance and Health (PAEPH) Research Group, Faculty of Sports Sciences, University of Extremadura, 10003 Cáceres, Spain; 2Social Impact and Innovation in Health (InHEALTH), University of Extremadura, 10003 Cáceres, Spain; 3Promoting a Healthy Society Research Group (PHeSO), Faculty of Sport Sciences, University of Extremadura, 10003 Cáceres, Spain; 4Social Impact and Innovation in Health (InHEALTH) Research Group, Faculty of Sport Sciences, University of Extremadura, 10003 Cáceres, Spain; 5Grupo AFySE, Investigación en Actividad Física y Salud Escolar, Escuela de Pedagogía en Educación Física, Facultad de Educación, Universidad de Las Américas, Santiago 8370040, Chile

**Keywords:** inclusion, attitudes, inclusive education, physical education, student, disabilities

## Abstract

Inclusive education for disabled people is becoming increasingly important globally. Improving the factors that support the inclusion of people with disabilities in education is one of the main objectives. In addition to teachers, another major factor is how the attitudes of students without disabilities affect those with disabilities, which should be considered in maintaining an inclusive classroom climate. The aim of the study was to analyse the attitudes of non-disabled students towards the inclusion disabled students in Physical Education (PE) and to investigate differences according to gender and school location. A total of 805 girls and boys participating in PE in public secondary schools (12–18 years old) in Extremadura were analysed through the AISDPE (Attitudes towards the Inclusion of Students with Disabilities in PE) questionnaire. The results show students without disabilities have positive attitudes towards the inclusion of students with disabilities. The majority were female. No significant differences were found regarding the location of the school. There are positive attitudes towards the inclusion of students with disabilities in the PE classroom, but these could be improved, especially in aspects more related to cognitive factors. For this, it is necessary for teachers to provide their students with the necessary tools and knowledge to better understand the possibilities and difficulties presented to students with disabilities, thus promoting a more inclusive classroom.

## 1. Introduction

Worldwide, there has recently been a significant movement regarding inclusion [1], one of the Sustainable Development Goals (SDGs) proposed by the United Nations (UN) for the 2030 Agenda and introduced in the Incheon Declaration that establishes the shaping of an inclusive, equitable, and quality education system [2,3]. The UN and the United Nations Educational, Scientific and Cultural Organizations (UNESCO) have taken the lead in encouraging governments, non-governmental organizations, development partners, civil society, and the media to advocate for inclusive education, issuing several declarations that highlight the need to educate all children, youth, and adults with special educational needs, such as the Salamanca Statement [4], the Dakar Framework for Action [5], and the Incheon Declaration [3] mentioned above.

For several years now, in many European countries, a large percentage of students with special educational needs [6] have been sharing their learning process with their peers in mainstream schools, thus reducing the number of special schools [7]. Along these lines, several authors have highlighted the importance of the attitudes of the actors most involved and who have the greatest degree of interaction with students with disabilities, such as teachers and non-disabled pupils [8,9,10]. Attitude is understood as a combination of beliefs, perceptions, and sensations in favour or of against in response to the educational situation [11].

On the one hand, teachers’ attitudes are very important for the development of an inclusive classroom, and teaching methodologies and strategies are important contribute significantly to this [12,13]. Several studies have been carried out in which teachers themselves discuss their perception of their initial and in-service training with respect to inclusive education, addressing their sense of competence and their capacity for inclusion [14,15,16,17]. However, there is currently some uncertainty about the attitudes of typically developing pupils toward their classmates with disabilities, with the moral identity development (cognitive readiness) and attitudes towards inclusive education (behavioural readiness) of these non-disabled students being potential determinants in the development of effective and high-quality inclusive education [18]. In relation to this, Ajzen and Fishbein’s theory of planned behaviour maintains that these attitudes have three components: affective, which involves the person’s feelings and emotions toward others; behavioural, which consists of actual or intended behaviour toward others; cognitive, which addresses the person’s knowledge about people with disabilities [19,20].

The negative attitudes of typically developing pupils about pupils with disabilities are a major barrier to increasing access to mainstream schools for pupils with disabilities [21]. Students with negative attitudes may feel uncomfortable or show little satisfaction when interacting with classmates with disabilities, showing negative body signals, avoiding eye contact with them, and possibly avoiding such interactions in the future [18,22]. These aspects may influence parents, who prefer to send their children with disabilities to special schools rather than mainstream schools and feel that their children are excluded by the negative attitudes shown by non-disabled peers in mainstream schools [23].

In the case of physical education, students with disabilities often experience social isolation, bullying, limited participation in physical activities, and even forced exclusion by their typically developing peers in the classroom [24,25]. Therefore, in order to develop inclusive PE, it is essential to create a favourable social environment full of positive attitudes towards students with disabilities by the social actors involved, namely, teachers, parents, and non-disabled classmates [26,27]. 

In this sense, the positive attitudes of non-disabled children, based on favourable behaviours and a greater willingness to play with peers who have a disability, could play a key role in promoting respect and acceptance towards this type of pupils [28,29]. Consequently, it has been shown that social interactions can improve the self-esteem, sense of socialization, and physical self-concept of students with disabilities in physical education classes [30], creating a favourable inclusive environment in which friendships can develop between typically developing classmates and classmates with disabilities [31].

To assess attitudes towards inclusive education, as mentioned above, much knowledge exists from the perspective of teachers and from the outlook of students with disabilities. However, there is less evidence on the attitude of typically developing peers toward inclusion, and even less so in the context of PE. Therefore, the present research aims to analyse the attitudes of students towards the inclusion of students with disabilities in PE and to investigate whether there are differences according to gender and school situation. On the other hand, we will also study the differences in the scores obtained in the two dimensions (cognitive and attitudinal) of the AISDPE questionnaire as a function of gender and school location.

## 2. Materials and Methods

### 2.1. Participants

A non-probabilistic method with convenience testing was applied to select the sample, which involved 805 students taking the subject of Physical Education in public secondary schools in Extremadura. According to the report of the European Foundation Society and Education for 2019 in Extremadura, the pupils enrolled in Obligatory Secondary Education (E.S.O) and the Baccalaureate were 59,814 (available at https://www.sociedadyeducacion.org/site/wp-content/uploads/Indicadores-comentados-2019.pdf, accessed on 15 September 2022). The sample size is 805 students from public schools in Extremadura, exceeding a confidence level of 95% with a margin of error of +5%. The selected courses ranged from the first year of Obligatory Secondary Education (E.S.O) to the second year of Baccalaureate School. Of the total sample, 46.3% (*n* = 373) were male and 53.7% (*n* = 432) were female. The mean age was 14.57 years. The following table (Table 1) characterises the participants sociodemographically.

### 2.2. Procedure

To access the sample, we consulted the guide to schools in Extremadura within Spain and selected all the schools that met the following inclusion criteria: they teach compulsory secondary education (12–16 years old) and high school (16–18 years old) and are public schools.

Once all the schools had been selected and the mail addresses of all of them had been obtained, an e-mail was sent to each school addressed to the teachers of the PE area. In the e-mail, the objective of the study was explained, and parental informed consent was provided, and it was explained to them that they could leave the study whenever they wanted to. Those schools that wished to collaborate in the study were required to collect informed parental consent for the students belonging to the classes that agreed to collaborate in the research and were required to make an appointment for the research team to visit the school and administer the questionnaires digitally through electronic tablets owned by the laboratory.

Once the research team went to the schools to administer the questionnaires during the physical education class in the presence of the teacher, a researcher provided each student with a tablet by which to access the form through a URL link and read aloud each item to ensure that all students understood the questionnaire. In addition, any doubts that may have arisen were resolved. The average response time for the questionnaire was 8 min.

It was decided to use an electronic questionnaire elaborated through the Google Forms application. All data were collected and processed anonymously between January 2022 and April 2022. The questionnaire was composed of four sociodemographic questions and the questionnaire on Attitudes towards the Inclusion of Students with Disabilities in PE (AISDPE).

This work was made in accordance with the directives of the Declaration of Helsinki and it was admitted by the “Comité de Bioética y Bioseguridad de la Universidad de Extremadura” (186/2021).

### 2.3. Instruments

Sociodemographic information: A sociodemographic questionnaire was prepared with five questions, namely, those pertaining to sex, age, weight, height, and level of education.

AISDPE: The AISDPE questionnaire was used to analyze attitudes towards the inclusion of students with disabilities in PE [32]. It is an amended version of the ATDQ (Attitudes Towards Disability Questionnaire). The original language is Spanish, so no cultural adaptation or translation was necessary. The procedure to include some items or others in each of the subscales was carried out through a confirmatory factor analysis. The instrument is composed of 17 items which are grouped into two dimensions (Table 2). Component 1 “Cognitive readiness of children with disabilities” consists of seven items and component 2 “Behavioural readiness to interact with children with disabilities” consists of ten items The questionnaire uses a Likert scale (1–5), with 1 being “strongly disagree”; 2, “strongly disagree”; 3, “indifferent”; 4, “strongly agree”; and 5, “strongly agree”. The authors reported a Cronbach’s alpha value of 0.82 for the cognitive component and 0.75 for the behavioural component. All parameters were reversed so that a higher value on the scale implied a higher degree of disagreement with the statement and therefore better attitudes towards inclusion. 

### 2.4. Statistical Analysis

First, to determine the kind of statistical analysis to be used, depending on whether the assumption of normality was met, the distribution of the data was explored using the Kolmogorov–Smirnov test. The result was that this assumption was not met, so it was decided to use nonparametric statistical tests. Secondly, the Mann–Whitney U test was applied to analyze the differences between each of the items of the AISDPE questionnaire and each of its dimensions as a function of the variables of sex and the location of the school. The Bonferroni correction was applied, so a significance level was established for *p* < 0.003. To analyse the relationship between each of the two factors of the instrument and the course variable, Spearman’s Rho test was used. Cronbach’s alpha was used to analyse the reliability of the instrument. To interpret the reliability values, reference was made to Nunnally [33], which showed that reliability scores of 0.70 until 0.90 can be satisfactory. Continuous variables are shown as the mean and standard deviation and categorical variables are presented as number and percentages. SPSS statistical software, version 23, for MAC was used for data processing.

## 3. Results

Table 3 presents the descriptive parameters of the questionnaire as a function of sex and school location. With respect to gender, it stands out that women scored higher on all items, obtaining statistically significant differences in all items except item 17 (*“If I became a wheelchair user due to an accident, my life would be meaningless”*). Concerning the location of the school, no statistically significant differences were found in any item; however, students belonging to rural schools scored higher on all items—except item 13 (*“People with disabilities are usually less intelligent than other people”*)—than students belonging to urban schools.

Table 4 presents the descriptive analysis in each component of the AISDPE questionnaire as a function of sex and school location. Statistically significant differences were obtained according to sex in both dimensions, with girls scoring higher than boys. No statistically significant differences were found according to school location in either of the two components, although students from rural schools obtained higher scores than students from urban schools.

Spearman’s Rho test was used to analyze the relationship between the cognitive and behavioural components of the AISDPE and the school grades (Table 5). No significant association was found.

Finally, Cronbach’s alpha was calculated to obtain the reliability indices for each of the dimensions of the AISDPE questionnaire. The results were as follows: a1 (cognitive perception for children with disabilities) = 0.772; a2 (behavioural readiness to interact with children with disabilities) = 0.809. These were considered satisfactory according to Nunally [33]

## 4. Discussion

The aim of the study was to examine the attitudes of non-disabled versus disabled pupils towards the inclusion of disabled students in PE and to observe differences according to gender and school location.

The findings of the present research show that female students show more positive attitudes towards the inclusion of students with disabilities in PE classes than male students, scoring higher on all items of the AISDPE questionnaire. The differences between females and males were statistically significant for all items, with the exception of item 17 (“If I were left in a wheelchair because of an accident, my life would be meaningless”), where the differences were not statistically significant. As for the school location, the differences were not statistically significant for any item of the AISDPE questionnaire, and the scores obtained between both types of school were quite similar, although they were slightly higher in rural schools for all items except for item 4, where the scores were equal, and item 13, where the urban school obtained a slightly higher score. Regarding the two dimensions of the AISDPE questionnaire, significant gender differences were also found in both dimensions, with females scoring higher than males in “cognitive perception for children with disabilities” and “behavioural readiness to interact with children with disabilities”. No significant differences were found between rural and urban students in these two dimensions; however, rural students scored slightly higher than urban students.

Several studies support these findings, using a common instrument in all of them, namely, the CATCH scale (Chedoke–McMaster Attitudes Toward Children with Handicaps) [34,35], whose original scale consists of 36 items reflecting three dimensions (affective, cognitive, and behavioural) [36], but current studies use a smaller 6-item version, which includes a behavioural component such as the AISDPE questionnaire, specific to the subject of PE, and an affective component, reflecting student’s feelings and emotions when interacting with children with disabilities [37,38,39]. The questionnaire items in the present research do not include an affective dimension as such, as this is included in some items of the other two dimensions; instead, the items include a cognitive dimension related to student’s thoughts about the possibilities and limitations of students with disabilities. As in this study, in all the studies cited above, girls show more positive behavioural attitudes towards interacting with students with disabilities than boys. There is not much evidence showing better attitudes on the part of boys; however, in Al-Kandari’s study, Kuwaiti girls showed less favourable attitudes of social distance and less derogatory beliefs than boys towards people with intellectual and developmental disabilities, although there is no consistent theoretical explanation for this difference [40].

In the specific case of PE classes, in the research by Abellán et al., the AISDPE questionnaire was also used, and the age of the participants was also similar, i.e., from 12 to 18 years old to this research; however, the Likert scale was inverted—that is, the higher the score, the more negative attitudes of the students towards inclusion. Girls showed better attitudes than boys and, in addition, the behavioural readiness of students to interact with students with disabilities was more positive than the cognitive perception of these students, findings similar to those of the present study [41]. Towsend and Hassal, in their study on students with intellectual disabilities, maintained that this gender difference and the less positive cognitive conception for students with disabilities could be due to the competitive nature of boys, who cognitively perceive students with disabilities as less competent and show less receptive attitudes towards them; this was true in our study, where cognitive perception of students with disabilities was more negative in boys [42]. In this line, non-disabled students in the study by Ocete et al. (2022) expressed that the inclusion of peers with disabilities would help them to be better people; however, they considered the competitive factor as an important barrier to the inclusion of peers with disabilities, as they were more aggressive in order to win [43]. In relation to this, girls could be more receptive because of a possible inexperience in the sport [25] or because of their more empathic nature [44], prioritizing more benevolent behaviours and interaction with their peers.

In terms of school location, no significant differences were found in the present study, although students in rural schools showed slightly more positive attitudes towards inclusion than students in urban schools. In the study by Rojo-Ramos et al., the differences according to school location were statistically significant, showing a better attitude on the part of students in rural schools towards the inclusion of students with disabilities; however, the study was conducted with primary school students [45]. Currently, studies comparing attitudes towards the inclusion of students with disabilities between urban and rural students are scarce, although it was shown that students whose families have a lower socioeconomic status and attend schools with a less favourable environment show more positive attitudes towards inclusion [18,46].

### 4.1. Practical Applications

Overall, the results of the present research show a relatively positive attitude on the part of typically developing students toward students with disabilities; however, there is ample room for improvement and these attitudes could be even more favourable. In this sense, the PE departments of the centres and the teachers that comprise them should design disability awareness programs within the centres, offering an opportunity to raise awareness and improve attitudes towards people with disabilities and turning the PE classroom into an inclusive environment suitable for incorporating physical activities that promote the acceptance and inclusion of students with disabilities [25,47]. The use of adapted or inclusive sports in programming could be a very useful strategy, allowing students with disabilities to show their skills and abilities in these sports and to be perceived by their peers as more competent [48,49,50]. In addition, teachers should adjust their methodologies, using programs that encourage cooperative learning between typically developing students and students with disabilities, where priority is given to high-quality contact and bonding through interpersonal experiences, rather than frequent contact without a clear objective where such an interaction is felt negatively by both parties [51,52].

### 4.2. Limitations

It was not possible to establish cause–effect relationships as this was a cross-sectional study. Therefore, it would be interesting for future research to further explore these findings in order to establish causal relationships.

The sample of this research represents students in schools in Extremadura, which is a region in the southwest of Spain. Their cultural and social characteristics may influence the data analysed. In this sense, it would be interesting to be able to extend this work to other regions of Spain, obtaining representative findings in order to observe attitudes towards students with disabilities throughout Spain. Although there are some items that include a specific disability, most of them do not specify any disability, but refer to students with disabilities as a general concept. In future studies on inclusive classrooms in PE, it would be useful to assess these attitudes distinguishing between various disabilities, since some studies have reported that, for example, attitudes towards students with intellectual disabilities were less positive than towards students with physical disabilities [53,54].

## 5. Conclusions

This work demonstrates that attitudes towards students with disabilities in PE classrooms are positive, especially among girls. No differences were found between urban and rural schools. Cognitive attitudes are lower than behavioural attitudes. It would be interesting for teachers to provide their students with the necessary tools and knowledge to better understand the difficulties faced by students with disabilities, aiming for a more inclusive classroom.

## Figures and Tables

**Table 1 children-10-01008-t001:** Sociodemographic characteristics of participants (N = 805).

Variable	Categories	N	%
Sex	Male	373	46.3
	Female	432	53.7
School location	Rural	425	52.8
	Urban	380	47.2
Grade	1º E.S.O.	149	18.5
	2º E.S.O.	150	18.6
	3º. E.S.O.	221	27.5
	4º E.S.O.	194	24.1
	1º Baccalaureate	61	7.6
	2º Baccalaureate	30	3.7
**Variable**		M	SD
Age		14.57	1.47

N: number; %: percentage; SD: standard deviation; M: Mean; E.S.O.: Obligatory Secondary Education.

**Table 2 children-10-01008-t002:** Dimensions of the AISDPE questionnaire and the items they contain.

Factor	Description	Items
1	Cognitive readiness of children with disabilities	1, 3, 4, 6, 13, 14, and 15
2	Behavioural readiness for interacting with children with disabilities	2, 5, 7, 8, 9, 10, 11, 12, 16, and 17

**Table 3 children-10-01008-t003:** Descriptive parameters of the questionnaire as a function of sex and school location.

	Sex			School Location		
Item	Male	Female		Rural	Urban	
	M (SD)	M (SD)	*p*	M (SD)	M (SD)	*p*
	Cognitive readiness of children with disabilities					
1. I believe that people with disabilities have more difficulty than other people in achieving the same personal and/or professional goals.	2.92 (1.30)	3.43 (1.11)	<0.001	3.24 (1.27)	3.04 (1.20)	0.027
3. I will stand out if I participate with people with disabilities in physical activities or sports.	3.18 (1.26)	4.02 (1.06)	<0.001	3.68 (1.23)	3.58 (1.22)	0.202
4. People with blindness should always be assisted by a guide.	2.35 (1.16)	2.76 (1.23)	<0.001	2.57 (1.19)	2.57 (1.24)	0.889
6. I would not want the teacher to tell me that I have to help a person with a disability.	3.69 (1.20)	4.60 (0.85)	<0.001	4.21 (1.12)	4.14 (1.13)	0.336
13. People with disabilities tend to be less intelligent than others.	3.72 (1.08)	4.44 (0.81)	<0.001	4.10 (1.02)	4.11 (1.00)	0.885
14. In general, people with disabilities are less sociable.	3.56 (1.13)	3.97 (1.01)	<0.001	3.86 (1.03)	3.69 (1.14)	0.055
15. Many people with disabilities are unable to take care of themselves.	3.01 (1.24)	3.27 (1.12)	<0.001	3.15 (1.17)	3.14 (1.20)	0.969
	Behavioural readiness for interacting with children with disabilities					
2. People with disabilities are unable to adapt to a competitive environment	3.93 (1.19)	4.37 (0.86)	<0.001	4.24 (0.98)	4.09 (1.11)	0.092
5. Students with disabilities should not participate in regular Physical Education classes because they may be detrimental to the progress of their classmates.	4.36 (1.02)	4.78 (0.60)	<0.001	4.64 (0.81)	4.52 (0.88)	0.024
7. I prefer not to interact with people with disabilities	4.20 (1.13)	4.75 (0.61)	<0.001	4.53 (0.90)	4.46 (0.97)	0.418
8. If I have a family member with a disability, I will avoid talking about it with other people.	3.96 (1.24)	4.58 (0.81)	<0.001	4.33 (1.05)	4.26 (1.11)	0.367
9. I would not sit in class next to a classmate with a disability.	4.49 (0.90)	4.85 (0.49)	<0.001	4.74 (0.68)	4.63 (0.78)	0.018
10. I would not choose a teammate with a disability for my team.	4.01 (1.15)	4.53 (0.79)	<0.001	4.32 (0.99)	4.25 (1.02)	0.362
11. I would not participate as a volunteer in a camp for people with disabilities where I would have to help them with showering, meals, etc.	3.53 (1.31)	4.18 (0.99)	<0.001	3.93 (1.16)	3.82 (1.23)	0.204
12. If I had a disability, my lifestyle would change completely.	2.51 (1.25)	2.78 (1.19)	<0.001	2.76 (1.25)	2.54 (1.19)	0.015
16. People with disabilities should practice specific and independent sports	3.60 (1.26)	4.08 (1.02)	<0.001	3.91 (1.15)	3.80 (1.18)	0.182
17. If I became a wheelchair user due to an accident, my life would be meaningless.	3.82 (1.24)	4 (1.08)	0.085	3.96 (1.15)	3.88 (1.17)	0.248

*p* is significant < 0.003. M = mean value; SD = Standard deviation.

**Table 4 children-10-01008-t004:** Descriptive analysis in each component of the AISDPE questionnaire as a function of sex and school location.

	Total	Sex			School Location		
Dimensions	M (SD)	Male	Female	*p*	Rural	Urban	*p*
Cognitive readiness of children with disabilities	3.50 (0.75)	3.19 (0.78)	3.78 (0.60)	<0.001	3.54 (0.74)	3.46 (0.75)	0.191
2. Behavioural readiness for interacting with children with disabilities	4.08 (0.63)	3.84 (0.71)	4.29 (0.47)	<0.001	4.13 (0.63)	4.02 (0.64)	0.008

*p* is significant < 0.003. M = mean value; SD = Standard deviation.

**Table 5 children-10-01008-t005:** Dimensions’ correlation with grade.

Dimensions	Grade *ρ (p)*
1. Cognitive readiness for children with disabilities	−0.051 (0.145)
2. Behavioural readiness for interacting with children with disabilities	−0.014 (0.696)

The correlation is significant at the *p* < 0.05.

## Data Availability

Not applicable.

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
