# Peer review of "Attitudes of Non-Disabled Pupils towards Disabled Pupils to Promote Inclusion in the Physical Education Classroom"

_children, 2023, doi:10.3390/children10061008_

Round 1

Reviewer 1 Report

First, this reviewer would like to acknowledge and thank the authors of this manuscript for their consistently respectful and intentional focus on how perceptions impact reality. Second, I am impressed at the construction of this using the English language when presumably English is not the primary language of the authors. I am humbled by your abilities.  I would ask, however, that a thorough "copy edit" be applied to this manuscript as there do exist some grammatical issues. I will highlight 3 examples:

(1) Line 149: "slevel" was used when "level" is the term they were looking for.

(2) Line 218: Start a sentence with a capital letter.

(3) Line 229: A symbol was inserted, randomly, after work #8.

(4) Line 260: Include year in the citation.

Moving on, this reviewer does wonder why, within the demographic portion of the survey of the respondents, why the elements of height and weight were included.  This is particularly curious when those two metrics were not revisited ever again.  I also wonder if there were any associations between age and/or level of education on the perceptions toward students with disability?  

Then, this reviewer really appreciated the aspect of socio-economic status (SES) as a potential contributing factor on attitudes and perceptions.

Overall, this was a relatively unique perspective (of students without disability) toward their peers with disability.

Again, this was easy to read and I commend the authors on their skills in working with a new language.

Author Response

REVIEWER 1

First, this reviewer would like to acknowledge and thank the authors of this manuscript for their consistently respectful and intentional focus on how perceptions impact reality. Second, I am impressed at the construction of this using the English language when presumably English is not the primary language of the authors. I am humbled by your abilities.  I would ask, however, that a thorough "copy edit" be applied to this manuscript as there do exist some grammatical issues. I will highlight 3 examples:

Authors’ final response: Thank you very much for your appreciation. Sincerely, thanks to you the quality of the manuscript has improved.

(1) Line 149: "slevel" was used when "level" is the term they were looking for.

Authors’ final response: Thanks, it was modified.

(2) Line 218: Start a sentence with a capital letter.

Authors’ final response: You have right, thanks.

(3) Line 229: A symbol was inserted, randomly, after work #8.

Authors’ final response: Thank a lot. It was a mistake.

(4) Line 260: Include year in the citation.

Authors’ final response: Thanks, it was modified.

Moving on, this reviewer does wonder why, within the demographic portion of the survey of the respondents, why the elements of height and weight were included.  This is particularly curious when those two metrics were not revisited ever again. 

Authors’ final response: Thank you very much, this data (height and weight) was taken into consideration simply as an informative piece of information to allow us to characterise the sample.

Dimensions

Age ρ (p)

1. Cognitive readiness for children with disabilities

2. Behavioral readiness for interacting with children with disabilities

-0.035 (0.291)

-0.055 (0.098)

Dimensions

Educational levelρ (p)

1. Cognitive readiness for children with disabilities

2. Behavioral readiness for interacting with children with disabilities

-0.063 (0.078)

-0.001 (0.988)

I also wonder if there were any associations between age and/or level of education on the perceptions toward students with disability?  

Authors’ final response: Thank you very much, this analysis was carried out but no association was obtained, so we did not consider including this type of information in the article. However, if you adhere to it, we would be happy to provide this information.

Then, this reviewer really appreciated the aspect of socio-economic status (SES) as a potential contributing factor on attitudes and perceptions.

Overall, this was a relatively unique perspective (of students without disability) toward their peers with disability.

Authors’ final response: We are very grateful for your consideration and for your contribution to the improvement of the manuscript.

Reviewer 2 Report

Dear Authors

First of all, I would to congratulate you on the quality of the study in theoretical and methodological terms. However, it is suggested to review the concept of inclusion that you have used, since the current trend is to overcome the distinction "disability", to understand rather the diversity of the scholl classrom, in terms of gender, race, culture, religion, SEN or other dimensions that allow a more complex understanding of the concept of inclusion.

Author Response

REVIEWER 2

Dear Authors

First of all, I would to congratulate you on the quality of the study in theoretical and methodological terms. However, it is suggested to review the concept of inclusion that you have used, since the current trend is to overcome the distinction "disability", to understand rather the diversity of the school classroom, in terms of gender, race, culture, religion, SEN or other dimensions that allow a more complex understanding of the concept of inclusion.

Authors’ response: Thank you very much for your appreciation. We have reviewed the concept of inclusion as you have considered it. And we agree with you. Now, as you say, the concept of diversity in the classroom is more widespread. You are right, but perhaps the English language simplifies the concepts and we understood that it was easier to reach everyone with a universal language.